# Antibacterial Profile of a Microbicidal Agent Targeting Tyrosine Phosphatases and Redox Thiols, Novel Drug Targets

**DOI:** 10.3390/antibiotics10111310

**Published:** 2021-10-27

**Authors:** Kylie White, Gina Nicoletti, Hugh Cornell

**Affiliations:** STEM College, RMIT University, Melbourne, VIC 3001, Australia; ambrogina.nicoletti@rmit.edu.au (G.N.); humarg1@hotmail.com (H.C.)

**Keywords:** antimicrobial, nitropropenyl benzodioxole, nitroalkenyl benzenes, tyrosine signaling, protein tyrosine phosphatase inhibitor, thiol oxidant, redox signaling, drug target

## Abstract

The activity profile of a protein tyrosine phosphatase (PTP) inhibitor and redox thiol oxidant, nitropropenyl benzodioxole (NPBD), was investigated across a broad range of bacterial species. In vitro assays assessed inhibitory and lethal activity patterns, the induction of drug variants on long term exposure, the inhibitory interactions of NPBD with antibiotics, and the effect of plasma proteins and redox thiols on activity. A literature review indicates the complexity of PTP and redox signaling and suggests likely metabolic targets. NPBD was broadly bactericidal to pathogens of the skin, respiratory, urogenital and intestinal tracts. It was effective against antibiotic resistant strains and slowly replicating and dormant cells. NPBD did not induce resistant or drug-tolerant phenotypes and showed low cross reactivity with antibiotics in synergy assays. Binding to plasma proteins indicated lowered in-vitro bioavailability and reduction of bactericidal activity in the presence of thiols confirmed the contribution of thiol oxidation and oxidative stress to lethality. This report presents a broad evaluation of the antibacterial effect of PTP inhibition and redox thiol oxidation, illustrates the functional diversity of bacterial PTPs and redox thiols, and supports their consideration as novel targets for antimicrobial drug development. NPBD is a dual mechanism agent with an activity profile which supports consideration of tyrosine phosphatases and bacterial antioxidant systems as promising targets for drug development.

## 1. Introduction

Cellular signal transduction networks sense and transmit internal and external signals, resulting in coordinated responses to stimuli. Reversible phosphorylation on serine (Ser), threonine (Thr) and tyrosine (Tyr) residues in proteins is a major post translational modification that regulates signal-transduction and protein functions in cellular physiological processes [1]. Bacterial regulatory networks of kinases phosphatases are complex and interconnected and enable adaptation to the challenges of stressful and changing environments. Coordination between activating kinases, which control the amplitude of a signal response, and terminating phosphatases, which control response rate and duration, maintain cell homeostasis [2]. Protein tyrosine kinases (PTKs) and protein tyrosine phosphatases (PTPs) are important regulators of signaling events and are widely and heterogeneously distributed across eukaryotic and prokaryotic cells [3]. Bacterial PTKs and PTPs have diversified structurally and functionally from those in eukaryotes, performing differing and often unique functions making them suitable targets for selective inhibition [3].

Bacteria have unique auto-phosphorylating tyrosine kinases (BY-Ks) which govern metabolic functions including the cell cycle, DNA metabolism, transcription and gene expression. BY-Ks have relaxed substrate specificity, phosphorylate multiple proteins and are suited for adaptation to new environments. Bacteria have eukaryotic-type serine and threonine kinases (eSTK), that can phosphorylate on tyrosine and cross-phosphorylate with BY-ks [4].

Cysteine-dependent PTPs belong to structurally and functionally diverse families that share an essential and invariant cysteine residue in the catalytic motif. PTPs are classified by sequence differences in the catalytic and flanking domains which determine substrate specificity and enzyme functionality [2]. PTPs include tyrosine-specific PTPs (sPTP), low molecular weight PTP (LMWPTP) and dual-specific phosphatases (DSP) which can dephosphorylate Ser, Thr and Tyr residues. LMWPTPs, predominant in bacteria, are concerned with BY-K modulation and have a more diverse range of functions than eukaryotic LMWPTP [3].

Catalytic cysteine thiolates are highly susceptible to transient, reversible oxidation by nucleophiles and electrophiles which inactivate phosphatase function, and to reduction by cellular redox thiols and reductases, which restore function, ensuring efficient enzyme recycling. Redox regulation of PTPs indirectly regulates PTK signaling, PTP activation terminating, inactivation prolonging, and tyrosine signaling. Oxidation of non-catalytic cysteine residues in PTKs and PTPs also regulates their activity [5,6]. Reactive oxygen species (ROS) transiently oxidize and inactivate PTPs and other cysteine-dependent enzymes. Low molecular weight redox active thiols (LMWT) and enzymes such as thioredoxins and glutaredoxins play major roles in reversing the oxidative inactivation of enzymes, maintaining a reduced cytoplasm [7].

Phosphotyrosine (PTyr) proteins vary greatly in abundance in bacterial species and are involved in nucleotide metabolism, transcription, cell integrity, synthesis and transport of extracellular polysaccharides, stress responses and virulence activities [8,9]. Many bacterial PTPs are identified genomically or have functions inferred from homology with characterized PTPs. Their effects are largely identified by phenotypic changes in null mutants or chemically suppressed cells. Suppression by PTP inhibitors is often the first indication that PTPs play a role in a physiological function. PTPs positively and negatively regulate components of signal pathways and their inhibition could result in desirable or undesirable physiological effects, which might differ between species. They are present at very low levels and are transiently and variably active, depending on the environmental context, making identification of functions difficult [10].

Nitropropenyl benzodioxole (NPBD) is a lipophilic, cell permeable, neutral tyrosine mimetic, belonging to the family of nitroalkenyl benzenes (NAB) (Figure 1; Appendix A). Nitroethenyl benzene (NEB), nitropropenyl benzene (NPB), nitroethenyl benzodioxole and NPBD are reversible inhibitors of enzymatic activity of PTP1B, SHP1, Yop and CD45 with differing levels of inhibition of enzymatic function ([11,12,13], Appendix A). These compounds compete with PTyr substrate proteins for binding to PTP active sites. The electrophilic nitropropenyl and nitroethenyl substituents inhibit enzyme function through oxidation of the cysteine residue. The presence of mercaptoethanol strongly reduces NAB inhibition of enzymatic activity of PTP1B and SHP1 [11]. NABs are strong electrophiles and oxidants and readily undergo reduction by reversible Michael addition of nucleophilic thiolates [14]. Depletion of the redox thiol pool, which maintains a reduced cytoplasm and cytosol, results in increased levels of redox-reactive species and increased oxidative stress (OS), contributing to cell death [7]. Conversely, thiol reduction of the nitroalkene moiety of NABs suppresses antimicrobial activity, the nitro alkane substituted NABs being inactive [15,16].

NAB analogues with varied benzene substituents show different activity patterns for bacteria and fungi and different zebrafish embryogenic toxicities, indicating the significance of substituents on the benzene ring for substrate selectivity [3]. NPBD 0.2 mg/L inhibits zebrafish egg hatching and in embryos reduces the heartbeat rate and affects epiboly movement and eye development in a dose-dependent manner but does not cause morphological abnormalities [15]. Many antibiotics (0.001–0.2 mg/L) show a variety of toxic effects on zebrafish development and metabolism [17]. Zebrafish toxicity was not predictive of oral animal toxicity. NPBD has low oral toxicity and low absorption from oral administration in rodents and is well tolerated on IV administration ([15]; Appendix A). NPBD is a broad-spectrum agent. It is fungicidal, inhibits unicellular protozoan pathogens in vitro and in vivo ([15], Nicoletti unpublished data) and is selectively toxic to lung cancer cells ([18], Appendix A). NPBD (BDM-I), a broadly active NAB analogue, is in preclinical development as an antimicrobial drug for human therapeutic uses by Opal Biosciences Ltd.

We report here on the activity profile of NPBD against phylogenetically diverse bacterial species, providing a broad-spectrum evaluation of a selective PTP inhibitor which highlights the diversity of PTP functions in bacterial physiology, and identifying possible PTP targets in bacterial species. The report supports the proposal that tyrosine phosphatases and redox thiols are potential bacterial drug targets.

## 2. Results and Discussion

### 2.1. NPBD Is Broadly Active against Clinically Significant Bacterial Species

The MIC and MBC of NPBD and positive control Ciprofloxacin were determined across 11 orders comprising 16 families and 39 species (Table 1, Appendix A). The grouping of species by phylogenetic types facilitated detection of different activity patterns related to evolutionary diversity in structure and function.

NPBD showed broad but variable activity against Gram-positive bacteria (average MIC 10.5 ± 4.7 mg/L, range 1–49 mg/L), and was bactericidal at titres ≤2× MIC for all but staphylococci and enterococci. The *Clostridiales,* including *C. difficile* and *C. perfringens*, were the most susceptible (MIC range 1–8 mg/L). NPBD showed greater variability against Gram-negative pathogens, MIC titres ranging from 0.125 to >512 mg/L, however, all species, including those with MBC > 512 mg/L, had bactericidal titres ≤4× MIC. Gram-negative species with lipo-oligosaccharide (LOS)-type cell envelopes colonizing non-enteric mucosal surfaces were generally susceptible (24.5 ± 4.9 mg/L, range 0.125–128 mg/L). Within this group, *Pasteurella, Haemophilus and Neisseria* were highly susceptible (average 1.9 ± 1.9 mg/L) and *Brucella* and *Acinetobacter* moderately susceptible (42 ± 17 mg/L and 128 ± 0 mg/L respectively). The susceptibility of LOS-type enteric commensals varied. Ten species of *Campylobacter* had uniform MIC (183 ± 47 mg/L) suggesting a corresponding uniformity in PTP targets. *Y. enterocolitica* (16 ± 0 mg/L) and *Bacteroides fragilis* (23 ± 14 mg/L) were moderately susceptible. *Proteus* species (*Morganellaceae*) showed variability between strains (76 ± 30 mg/L), and NPBD was bactericidal to *P. vulgaris* but not to *P. mirabilis*. This is perhaps not surprising considering the confusing taxonomy of the *Proteus*, *Providencia*, *Morganella* group and their biochemical differences [19]. The *Enterobacteriaceae* and *Pseudomonadaceae* with LPS rich envelopes were the least susceptible group with MIC ≥512 mg/L. Six species which tolerated 5% DMSO well had average MBC ≤ 4× MIC, indicating a bactericidal effect (Table 1). These saprophytic species are adapted to complex environments and encode multiple membrane transporter proteins which export a wide range of xenobiotics, while intracellular and commensal species have limited and more specialized systems [20]. Efficient export may explain the low efficacy of NPBD against the *Enterobacteriaceae* and *Pseudomonadaceae.* The higher activity against LOS-type enteric species is unlikely to be due to penetrability. NPBD was equally effective against a *C. jejuni* LOS deficient mutant and the parent strain. NPBD does not alter membrane permeability of *E. coli* or damage spheroplasts of *E. coli* and *M. catarrhalis* [12]. NPBD MIC/MBC for antibiotic-resistant clinical isolates of *S. aureus*, *E. faecalis*, *E. faecium*, *P. mirabilis*, *E. coli* and *C. jejuni* were similar to those of laboratory strains suggesting no cross resistance to the classes of antibiotics tested (Table 1). *S. aureus* strains were resistant to three or more antibiotics, including nine resistant to methicillin, and all 4-fold to 16-fold more resistant to ciprofloxacin than ATCC 29213 (Appendix A). Ciprofloxacin showed high and uniform activity across all species and MIC ranges tor type strains were within CLSI and EUCAST ranges. The activity pattern indicates broadly distributed and highly susceptible target(s) (Appendix A).

In summary, NPBD showed high to moderate activity against Gram-positive and more variable and selective activity against Gram-negative species. It is active against *Staphylococcus* and *Enterococcus* species which cause problematic antibiotic-resistant infections. It is bactericidal to enteric pathogens *C. difficile*, *C. perfringens*, *Y. enterocolitica* and *Campylobacter* spp., to urogenital pathogens *N. gonorrhoeae*. and to respiratory tract pathogens *H. influenzae*, *S. pneumoniae, S. pyogenes*, *M. catarrhalis*, *N. meningitidis* and *Acinetobacter species.* It inhibits urogenital pathogens *Chlamydia trachomatis* [12] and *Candida albicans* [15] and respiratory pathogen *M. tuberculosis (*Appendix A*).* For skin and mucosal infections where higher concentrations are achievable it could be an effective antimicrobial agent. NPBD also inhibits virulence factors in vitro; prodigiosin production in *Serratia marcescens*, cell adherence and invasion by *Y. enterocolytica* and motility in *Proteus* spp. [12]. Inhibition of virulence factors would contribute to infection control at skin and mucosal surfaces.

Many species naturally produce phenotypically variant sub-populations which survive in response to stresses and which, on culture, manifest as slow growing small colony variants (SCV). Variants show diverse transient or stable metabolic changes, such as auxotrophies and defective electron transport, and, under non-stress conditions, can revert to the parental or to different distinct phenotypes [21]. NPBD at concentrations above the MIC induced the emergence of tolerant sub-populations in *S. aureus*, manifesting as SCV that were phenotypically unstable and not intrinsically resistant to NPBD, their prevalence depending on the presence NPBD (Appendix A).

The growth inhibitory activity of NPBD reported here indicates that it is targeting PTP involved in primary metabolism which, alone or in combination, are necessary to viability, cell cycle progression or cell division. Tyrosine phosphatases play significant roles in bacterial responses to environmental stresses and in activities which enable infection and protect against host defences [8]. Tyrosine signaling also interacts with bacterial second messenger, cyclic di-guanosine monophosphate, and quorum-sensing auto-inducers in virulence activities [8,22]. NPBD inhibits many virulence factors in bacteria which involve PTP signaling. Although important to the activity profile of a PTP inhibitor, these are not of immediate relevance to explaining the effects of NPBD presented in this paper [12] and Nicoletti unpublished.

Few bacterial PTPs have been characterized and most are linked to species-specific functions, many of relevance to infectivity. They provide examples of likely targets for inhibition by a PTP inhibitor. *S. aureus* has the non-essential LMWPTPs, PtpB and PtpA, the latter being important to survival and infectivity [23]. *B subtilis* LMWPTP, YwlE, and Yfkj are involved in stress resistance [24]. A BY-K, PtkA with no identified cognate PTP is involved in DNA replication in *B. subtilis* [25]. Two putative PTKs and one PTP similar to proteins in *B. subtilis* and *S. pneumoniae* are reported in *E. faecium* [26]. *S. pyogenes* encodes no BY-K but has a DSP, SP-PTP, which positively regulates growth, cell division and expression of virulence genes. LMWPTP Spd1837 in *S pneumoniae* is involved in virulence but not capsule production [27]. *Y. enterocolitica* YopH is essential to virulence [8]. NPBD inhibits YopH enzymatic activity and adhesion and invasion of human cells [12].

Of more significance to growth inhibition are PTP reported to be involved in many aspects of primary metabolism. Complex cross-phosphorylation between STKs, BY-Ks and phosphatases are involved in DNA metabolism, transcription, cell division and sporulation in *B. subtilis* [28]. Phosphorylation on Ser, Thr and Tyr regulate binding proteins that modulate DNA repair and replication in *B. subtilis* and *E. coli* [29]. PTPs regulate binding of transcription factors to DNA in *B. subtilis, S. pneumoniae, S. aureus and K. pneumoniae* [30]. Tyrosine phosphorylation negatively regulates the β-subunit of RNA polymerase (RpoB), which modulates transcription in *S.*
*pneumoniae*, *Helicobacter pylori* and *K. pneumoniae* [31]. In *B. subtilis* PtkA, PtkB and PtpZ regulate RpoB [28]. Tyrosine replacement in the RpoB binding site in *E. coli* decreases RpoB affinity for DNA, suppressing capsular polysaccharide production [32]. The ultimate effect of inhibition of PTPs negatively regulating rpoB will depend on the functions of expressed proteins. NPBD suppresses endospore formation in *B. subtilis* and down-regulates RNA binding protein SpoVG which is associated with cell division and initiation of sporulation [12,33]. Deletion of *SpoVG* in *S. aureus* reduces methicillin and vancomycin resistance [34]. NPBD was active against VRE and MRSA strains (Table 1) and may inhibit a PTP positively regulating SpoVG expression. *Mycobacterium tuberculosis* PtpA positively regulates human and mycobacterial ATP synthase α-subunit AtpA, resulting in down-regulation of AtpA, reduced ATP synthesis and reduced growth. Orthovanadate inhibits PtpA resulting in down-regulation of AtpA, reduced ATP synthesis and reduced growth [35]. Mtb is sensitive to high ROS levels and has difficulty maintaining redox homeostasis [36]. NPBD inhibits the growth of Mtb in vitro ([37], Appendix A). The above examples illustrate the complexity of tyrosine signaling in bacteria and the positive and negative effects of inhibition of particular PTPs. The physiological result of exposure to an inhibitor indicates the balance of inhibition or promotions of all susceptible PTP. The functional diversity of PTPs from those in higher organisms suggests that inhibition can be selective. As more PTPs are isolated and characterized a more co-ordinated understanding of the complex role of tyrosine signaling in bacteria will emerge. The value of selective PTP inhibitors is to identify new physiological functions which involve PTP.

### 2.2. Plasma Binds NPBD, Lowering the Bioavailability of NPBD

Protein binding by covalent, electrostatic or hydrogen bonding affects the pharmacokinetics and pharmacodynamics of a drug, reducing the level of free drug in blood and tissues and affecting the efficacy of antibiotics [38]. NPBD binds strongly and reversibly (84% ± 4.2) to human serum albumin (HSA), the predominant redox thiol in plasma (Appendix A). The effect on antibacterial activity of binding of NPBD to blood proteins was estimated by determination of MIC/MBC titres in the presence of plasma. Antibacterial activity was reduced in a dose dependent manner in the presence of increasing plasma levels. A 50% plasma concentration increased 32-fold the MIC for *S. aureus* and *S. pyogenes* and the MBC for *S. pyogenes* (Appendix A). This data suggests the dosing required for inhibition of staphylococci and streptococci in blood and tissues would be considerably higher than dosing based on the standard MIC titre. There are no established quantitative relationships between blood protein binding in vitro and in vivo but inhibition of antibacterial activity is a predictor of lowered bioavailability in blood and tissues [38]. HSA is the predominant redox thiol in plasma and its reduction of NPBD would lower bioavailability.

### 2.3. Thiol Reduction of NPBD Decreases Antimicrobial Activity

Low molecular weight thiols are nucleophiles and reductants acting as redox buffers to maintain a reduced cytosol and preserve thiol enzyme cofactors, Cys-dependent antioxidant enzymes and PTPs [7]. LMWT, present in highly reduced form in mM concentrations, vary in type and level across bacteria, fungi and unicellular protozoa to meet diverse oxidative conditions. The major ROS scavengers in bacteria are glutathione (GSH) or the cysteinyl glycosides (bacillithiols or mycothiols), free cysteine, and Co-enzyme A [7,39].

The interaction of thiols with NPBD was investigated by the effect of cysteine and dithiothreitol on the MIC/MBC for *S. aureus*, *E. faecalis*, *B. subtilis* and *P. vulgaris*. Molar excess (1 and 10 mM) of cysteine and dithiothreitol increased the MIC for all species in a dose-dependent manner, 10 mM increased the MIC ≤ 2-fold for cysteine and ~20-fold for DTT (Table 2). The differing effects of thiol excess in the test species may reflect the species-specific nature of LMWT. Most Gram-positive bacteria, including *Staphylococcus* and *Bacillus,* use cysteine and bacillithiols. Gram-negative bacteria and some Gram-positive species, including *Enterococcus and Streptococcus,* use cysteine and GSH or mycothiols [39]. Actinobacteria, including *Corynebacterium* and *Mycobacterium*, use mycothiols. Reductive inactivation of NPBD in the presence of thiols confirmed that oxidation of thiols contributed selectively to disruption of bacterial cell redox balance thus contributing to oxidative stress and indirect repression of PTP function.

### 2.4. NPBD Shows Varying Bactericidal Action on Rapidly Growing, Slowly Growing and Non-Growing Bacteria

Investigation of the dynamic effect of NPBD under different growth conditions was assessed by time-kill (TK) assays. The relationship between agent concentration and its effect on population density over time is expressed as kill curves or as log_10_ reduction factors (RF) [40]. Patterns of bactericidal rates under different growth conditions can reveal agent effects on dormant and slow-growing cells and the presence of phenotypically variant sub-populations of agent-refractory cells, which may contribute to treatment failure and chronic infection [41].

The bactericidal activity of NPBD (2–8× MIC) was assessed for Gram-positive and Gram-negative species by TK assays in broth and in distilled water at room temperature where cells would be non-replicating. Population reduction patterns differed between species and for replicating and non-replicating cells, reflecting the variation in functions of targeted PTPs (Figure 2 and Figure 3).

NPBD showed rapid dose-dependent reduction of *C. xerosis* and *B. subtilis* in broth and of *B. subtilis* in water (Figure 2). The kill rates for LOS-type *M. catarrhalis* and *Y. enterocolitica* in broth and *A. calcoaceticus* in water were rapid and dose-dependent (Figure 3). NPBD reduced *P. vulgaris* cells rapidly in broth, with low-level dose-dependence, but did not kill cells in water. Dose-dependent suppression in broth and water suggests the inhibition of PTP(s) necessary for cell viability. Dose dependent reduction in broth but not water suggests affected PTP(s) are involved in replication but are not essential. NPBD showed slow, dose-independent population reduction of *S. aureus* and *E. faecalis* in broth and water (Figure 2). NPBD (2× MICi) was equally inhibitory to rapidly (RF 0.8) and slowly growing (RF 0.9) *S. aureus*, however, less reduction of non-metabolizing cells (RF 0.3), suggests NPBD affects replication and is truly bacteriostatic to staphylococci and enterococci (Figure 4). Exponentially replicating cells of *S. aureus* are rapidly killed by ‘bactericidal’ aminoglycosides, daptomycin, beta-lactams, quinolones and vancomycin (10–50× MIC), but only aminoglycosides and daptomycin are bactericidal to stationary phase populations [42].

### 2.5. NPBD Has Low Level Interactions with Bacteriostatic and Bactericidal Antibiotics

A comparison can be made, using a static MIC chequerboard titration, of the growth inhibitory effect of two agents in combination compared to either agent alone. When using drug concentrations achievable in vivo, this method provides efficacy data to guide combination therapy [43]. The chequerboard assay was here used to investigate interactions between NPBD and four antibiotics differing in mechanism and degree of activity against *S. aureus* and *E. faecalis*. Tetracycline and erythromycin interfere with different aspects of ribosomal function and are designated as ‘bacteristatic’ drugs for pharmacokinetic purposes. Vancomycin and ciprofloxacin are ‘bactericidal’. Vancomycin interferes with cell wall synthesis and alters membrane permeability. Ciprofloxacin interferes with DNA gyrase and topoisomerase, inhibiting separation of DNA strands and thus inhibiting cell division [44].

The concentration of each agent in inhibitory combinations, relative to the inhibitory concentration of each agent alone, is used to derive a fractional inhibitory concentration index (FICI), categorized by defined value ranges of pharmacodynamic usefulness: FICI <0.5 indicating synergy, >0.5 to <4 no significant interaction, and FICI of >4 indicates antagonism [43]. The FICI for *S. aureus* and *E. faecalis* for NPBD/antibiotic combinations indicated no interaction (Table 3).

The inhibitory potency of NPBD/antibiotic pairs was further characterized by summation of all FIC showing minimum (ΣFIC_min_) and maximum inhibition (∑FIC_max_) where a ΣFIC_max_ ≥ 2 is considered indicative of antagonism and ΣFIC_min_ < 0.75 of synergy [45]. NPBD did not modify the activity of ciprofloxacin, vancomycin or erythromycin for *S. aureus*, indicated by the small difference between ΣFIC_min_ and ΣFIC_max_. The result for tetracycline was close to the cut-off for synergism (ΣFIC_min_ 0.76). For *E. faecalis,* NPBD enhanced the activity of tetracycline (ΣFIC_min_ 0.19) and ciprofloxacin (ΣFIC_min_ 0.5) and was equal to the cut-off for synergism for vancomycin (ΣFIC_min_ 0.75) (Table 3). FIC indices assume that test drugs have similar linear dose-response curves whereas antibiotics generally show varying kill patterns [46].

The effect of NPBD on the bactericidal rate of tetracycline for *E. faecalis* was investigated by TK assays. NPBD (1×–16× MICi), showed a dose-independent, slow bactericidal effect compared to the dose-dependent kill rate of tetracycline (Figure 5a,b). Equal concentrations of NPBD and tetracycline (1×–16× MICi) had a lower dose-dependent reduction than tetracycline alone (Figure 5c).

Bacteriostatic drugs generally have antagonistic effect on the kill rates of bactericidal drugs and such effects are greater where different metabolic functions are targeted and where differences in growth rates are significant, suggesting the effect on growth dynamics is more significant than the difference in mechanism of action [46]. The lesser bactericidal activity of NPBD may explain its antagonistic effect on the kill rate of tetracycline. Tetracycline has a broad antimicrobial spectrum, inhibits RNA viruses and *Plasmodium falciparum*, and has many metabolic functions in human cells, but no involvement with tyrosine signaling is reported [47]. Many bactericidal antibiotics induce oxidative damage including ciprofloxacin, for which a marginal interaction was noted, and tetracycline [48]. A thiol oxidant potentiates the activity of isoniazid in *M. tuberculosis* [49]. NPBD may enhance efficacy by contributing to oxidative damage which could differ between antibiotics and the spectrum of redox thiols in species. *E. faecalis* contains the tripeptide GSH and *S. aureus*, a cysteinyl-glycoside (bacillithiol). NPBD oxidation of cysteine increased the MIC twofold for *E. faecalis* but not for *S. aureus*, suggesting a stronger redox reaction with GSH (Table 2).

### 2.6. NPBD Does Not Induce or Select for Resistant Variants in Antibiotic-Resistant Species

The ability of an agent to induce drug-resistant mutants or select for drug tolerance is relevant to the potential for development of clinical resistance. Drug tolerant cells are a common feature of bacterial populations in vivo, contributing to recurrent and chronic infections [21]. Antibiotic resistance arises through accumulation of mutations, both target-specific and related to general metabolic functions, which may alter susceptibility to other antibiotics [48]. Vancomycin-resistant *E. faecium* and *S. aureus* (VISA) show unstable mutations, altered metabolic functions and altered susceptibility to antibiotics [50]. Mutation in rpoB contributes to vancomycin resistance in MRSA, VISA and *E. faecium*. Rifampin-resistant MRSA with rpoB mutations show lower susceptibility to vancomycin [51].

Development of resistance in a strain is indicated by an irreversible increase in MIC that is greater than the accepted MIC titre range and is assessed in vitro by long-term exposure to sub-inhibitory drug concentrations. Genetic stability of mutants is tested by reassessment of the MIC after drug-free subculture. Drug-tolerant cells exhibit a sustained higher MIC which, on drug removal, reverts to the original strain susceptibility.

The development of resistance to NPBD by MRSA and VRE strains was assessed after exposure for 16 weeks with weekly monitoring of strain MIC. No strain developed stable resistant populations (Figure 6). Weekly MIC titres varied ≤4-fold and MBC titres were ≤2× MIC. MIC and MBC titres after drug-free passages differed ≤4-fold. There was no sustained rise in the MIC of any strain to suggest the presence of NPBD-tolerant cells. The low probability of multiple PTP mutations to resistance suggests emergence of resistance to NPBD would be delayed. Many antibiotics provoke resistance mutations in bacterial anti-oxidative enzymes [48]. NPBD has the advantage of dual mechanisms as an oxidant and PTP inhibitor in delaying the emergence of resistant strains. 

The complementary mechanisms of action of NPBD, selectively suppressing tyrosine phosphatases and disrupting redox balance in microbial cells, could make it an effective rapid broad-spectrum agent for clinically significant drug-resistant variants with limited therapeutic options [52]. Its activity profile suggests NPBD could be an effective drug for the treatment of skin and mucosal infections and tissue-localized infections where direct drug delivery is possible and higher concentrations achievable. The broad activity patterns of NPBD illustrate the diversity of function and distribution of PTPs in bacteria and is a significant contribution supporting the proposal that PTPs and redox thiols are selective targets for antibacterial drugs. Difficulties in the identification of selective, bioavailable, small molecule PTP inhibitors of validated human disease targets has delayed drug development [53]. A variety of similar inhibitor molecules are being investigated to treat human diseases such cancer, diabetes and obesity [54,55]. This report contributes to the attractiveness of development of antimicrobial PTP inhibitors. Given the greater functional diversity of microbial tyrosine phosphatases, such candidates are likely to be more successful than PTP inhibitors directed to cancer and other human diseases. Interference with redox balance is also gaining traction for direct inhibition or potentiation of existing antimicrobial drugs [56,57,58].

## 3. Materials and Methods

### 3.1. Chemicals, Reagents, Bacterial Strains

GMP NPBD 0.2M stock solutions in DMSO (AnalaR^®^, BDH Chemicals, Leicestershire, England); Ciprofloxacin (MP Biomedicals); Media, supplements, Anaerogen™, Campygen™ (Oxoid, Cambridge, UK). Yeast Nitrogen broth, Brucella broth (BBL™). Test strains obtained from ATCC, NCTC, RMIT Culture Collection. Clinical isolates from human pathology laboratories (Appendix A). A summary of the physicochemical properties of NPBD is provided in Appendix A.

### 3.2. MIC and MBC Broth Microdilution Assays

MIC/MBC were determined by CLSI broth microdilution [59,60]. All test systems contained 1% *v*/*v* DMSO. Reported MIC is lowest concentration showing no visible growth (100% inoculum inhibition). MIC and MBC are reported as geometric mean titres from all replicates. The effect of inoculum density on MIC was determined for assays and reported as MICi (Appendix A).

### 3.3. Time-Kill Assays

The MICi in CAMHB was determined for 8 representative bacterial strains. NPBD at multiples of the MICi was prepared in CAMHB prewarmed to 37 °C, or DW at RT, inoculated with log phase cultures to give final ~5 × 10^6^ cfu/mL, incubated, shaking, at 37 °C or RT and sampled at intervals to 24 h. Cell density was measured by viable count and expressed as time-kill curves and log_10_ RF. Assays were repeated two to four times and average counts reported.

### 3.4. Effect of NPBD on Population Growth under Optimal and Growth Limiting Conditions

Population growth was assessed in CAMHB (control) and YNB broth: YNB without additives (no growth); YNB with 0.01% *w*/*v* glucose and 0.0017% *w*/*v* Casamino acids (suboptimal growth); YNB with 0.5% *w*/*v* glucose and 1.7% *w*/*v* Casamino acids (optimal growth). NPBD test concentrations (0.5×, 1×, 2× MICi) were inoculated with final log-phase 1 × 10^6^ cfu/mL suspension, incubated at 37 °C and cell density measured as above.

### 3.5. NPBD Interaction with Antibiotics 

NPBD (0.06–64× MIC), erythromycin, tetracycline, ciprofloxacin and vancomycin (0.01–16× MIC) log_2_ dilutions in CAMHB were dispensed to microtitre plates with increasing concentrations of each drug on adjacent axes and inoculated with log-phase (1 × 10^6^ cfu/mL) test suspensions. Concentrations for each agent alone and in combinations for wells showing no visual growth were recorded as Fractional Inhibitory Concentrations (FIC). A FIC index (FICI) representing the sum of FICs of antibiotic (A) with NPBD (B) was calculated as:$$\Sigma \mathrm{FIC}={\mathrm{FIC}}_{A}+{\mathrm{FIC}}_{B}=\left(\frac{C_{A}}{{\mathrm{MIC}}_{A}\;}\right)+\left(\frac{C_{B}}{{\mathrm{MIC}}_{B}\;}\right)$$
where MIC_A_ and MIC_B_ are the MIC of each agent alone and C_A_ and C_B_ are the concentrations of each agent in effective combinations. FICI < 0.5 indicates synergy, FICI > 0.5 to <4 no interaction, and FICI > 4 antagonism [43]. The highest (ΣFIC_max_) and lowest (ΣFIC_min_) were reported to provide higher sensitivity in detecting interactions, ΣFIC_max_ ≥ 2 indicating antagonism and ΣFIC_min_ < 0.75 indicating synergism [45]. A time kill assay was performed as above for *E. faecalis* exposed to equal concentrations (8–256 mg/L in CAMHB + 1% DMSO) of NPBD and tetracycline sampled at 0, 4, 8 and 24 h.

### 3.6. Development of Resistance on Continuous Exposure to NPBD

Vancomycin resistant *E. faecalis* and MRSA *S. aureus* were exposed to NPBD for 16 weeks. Each strain (1–5 × 10^5^ cfu/mL final) was inoculated (W_0_) into a 1 mL log_2_ dilution set 1 of NPBD (256–0.5 mg/L in CAMHB), incubated at 37 °C aerobically and sub-cultured weekly to fresh dilution sets and weekly MIC (W_1_ to W_16_) recorded. Representative colonies from MBC_W16_ plates were sub-cultured 3× in drug-free medium. The W_0_, W_16_ and W_17_ MIC/MBC were measured by microbroth dilution [60]. Acquisition of resistance was accepted as a >4-fold increase in the MIC_W17_ compared to MIC_W0_.

## 4. Patents

Denisenko, P.P.; Sapronov, N.S.; Tarasenko, A.A. Antimicrobial and radioprotective compounds. US Pat 9,045,452, 2 June 2015.[Claim: A method for the treatment of a gastrointestinal infection]Denisenko, P.P.; Sapronov, N.S.; Tarasenko, A.A. Antimicrobial and radioprotective compounds. US Pat 8,569,363, 29 October 2013.[Claim: A method for the therapeutic treatment of a skin or soft tissue infection]Denisenko, P.P.; Sapronov, N.S.; Tarasenko, A.A. Antimicrobial and radioprotective compounds. US Pat 7,825,145, 2 November 2010.[Claim: A method of treating vulvo-vaginitis]Nicoletti, A.; White, K. Protein tyrosine phosphatase modulators. WO/2008/061308, 29 May 2008.

## Figures and Tables

**Figure 1 antibiotics-10-01310-f001:**
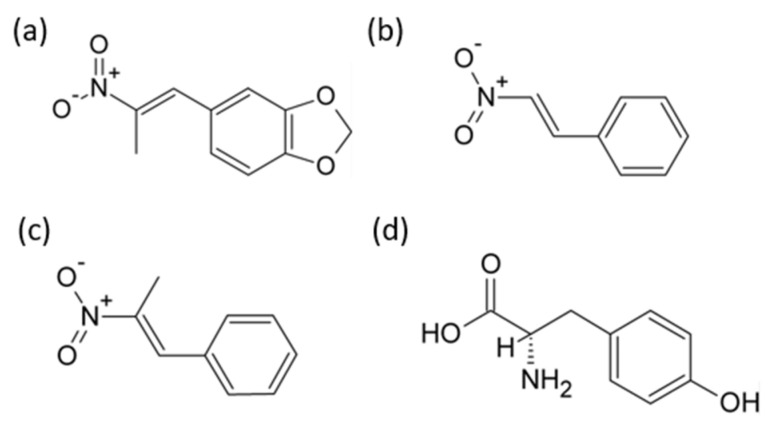
Structural similarity of PTP inhibitors (**a**) NPBD (5-(2-nitroprop-1-enyl)-1,3-benzodioxole), (**b**) NEB, (2-nitroethenylbenzene), and (**c**) NPB (2-nitroprop-1-enylbenzene) to (**d**) Tyrosine, ((2S)-2-amino-3-(4-hydroxyphenyl)propanoic acid).

**Figure 2 antibiotics-10-01310-f002:**
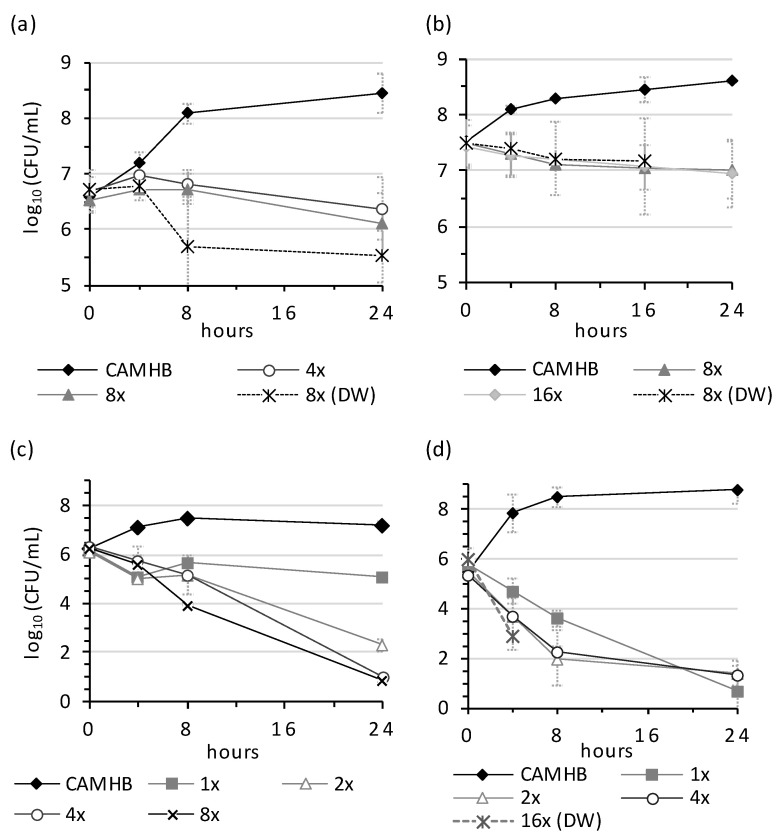
Bactericidal rates of NPBD for Gram-positive species. (**a**) *Staphylococcus aureus* ATCC29213, (**b**) *Corynebacterium xerosis* RMIT53/5, (**c**) *Enterococcus faecalis* ATCC29212, (**d**) *Bacillus subtilis* ATCC6633 in CAMHB or distilled water (DW). NPBD was tested at multiples of the MICi in CAMHB with 1% *v*/*v* DMSO. Limit of detection <10 cfu. Assays repeated 2–4 times and average viable counts reported.

**Figure 3 antibiotics-10-01310-f003:**
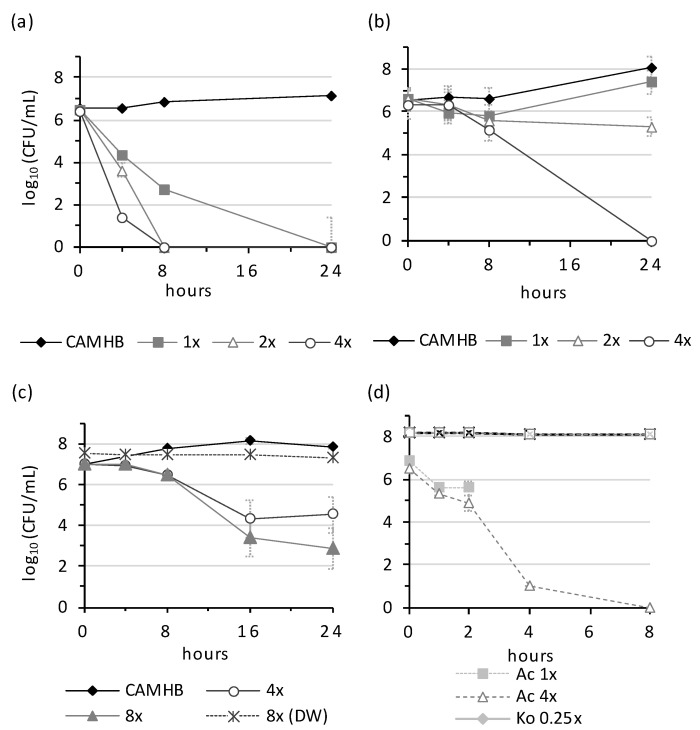
Bactericidal rates for NPBD against Gram-negative species. (**a**) *Moraxella catarrhalis* RMIT211/2, (**b**) *Yersinia enterocolitica* ATCC23715, (**c**) *Proteus vulgaris* ATCC13315 in CAMHB or distilled water (DW); and (**d**) *Acinetobacter calcoaceticus* RMIT3131 and *Klebsiella oxytoca* RMIT 180/4 in distilled water. NPBD was tested at multiples (×) of the MICi in CAMHB with 1% *v*/*v* DMSO. Limit of detection was <10 cfu. Assays repeated two to four times and average viable counts reported.

**Figure 4 antibiotics-10-01310-f004:**
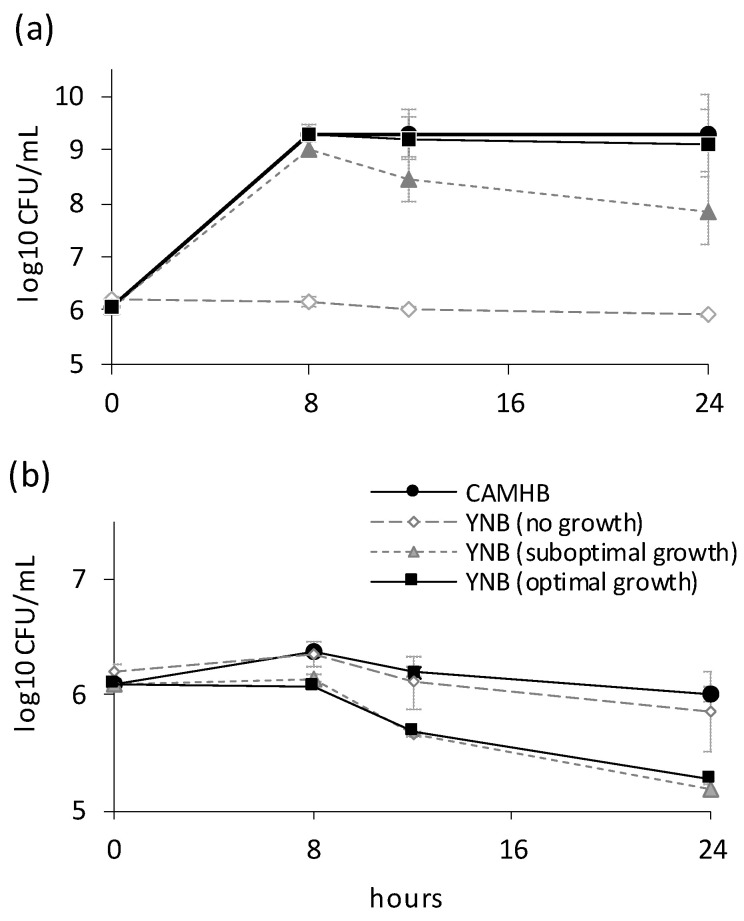
NPBD kills slow-growing cells of *S. aureus* at the same rate as rapidly growing cells. Rate of growth of *S. aureus* ATCC ATCC29213 in CAMHB and in media supporting no growth, suboptimal growth and optimal growth with (**a**) no NPBD and (**b**) 16 mg/L NPBD (2× MICi). Yeast Nitrogen Broth formulations: YNB without additives (no growth); YNB with 0.01% *w*/*v* glucose and 0.0017% *w*/*v* Casamino acids (suboptimal growth); YNB with 0.5% *w*/*v* glucose and 1.7% *w*/*v* Casamino acids (optimal growth). Final formulation contained 1% DMSO.

**Figure 5 antibiotics-10-01310-f005:**
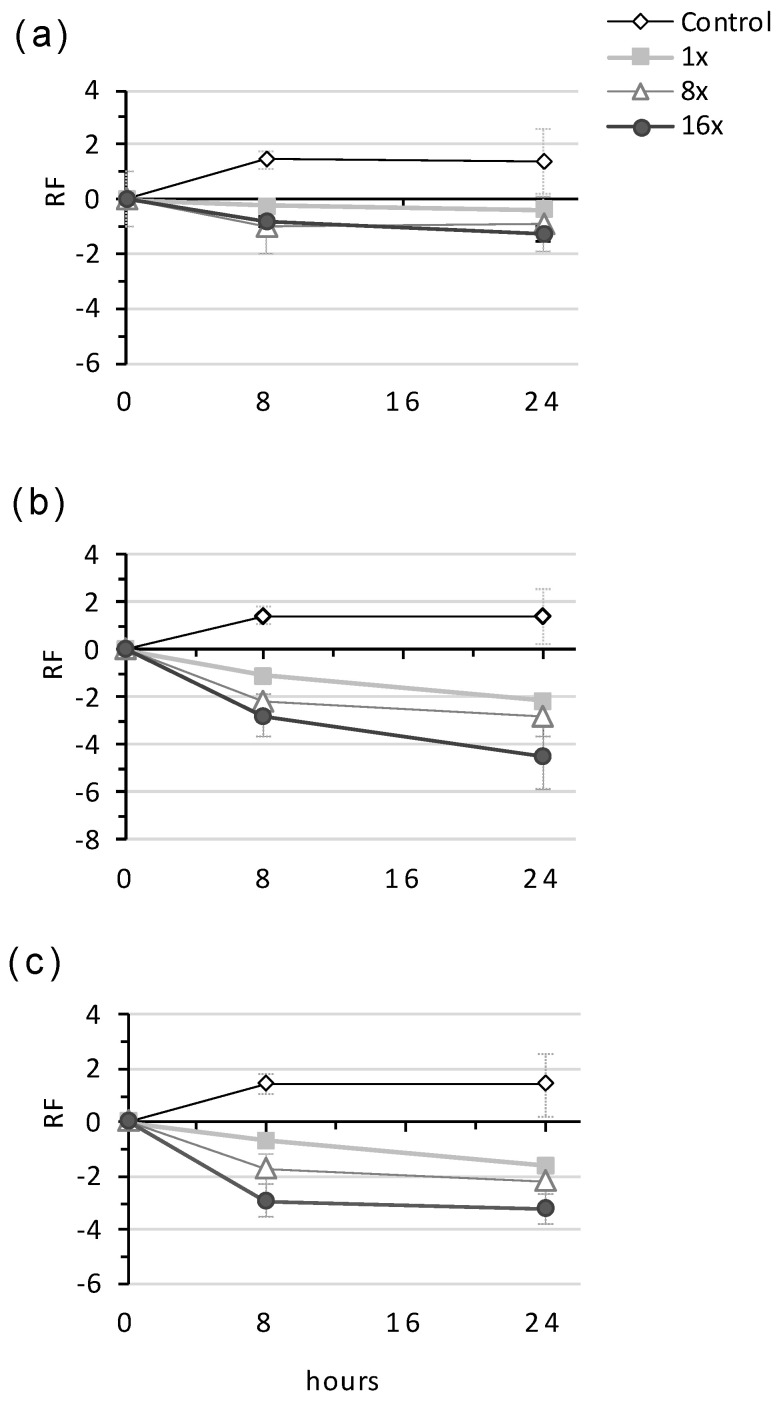
Antagonistic effect of NPBD on the action of tetracycline against *Enterococcus faecalis*. (**a**) NPBD alone; (**b**) Tetracycline (**c**) NPBD & tetracycline in equal proportions. MICi for tetracycline and NPBD was 16 mg/L.

**Figure 6 antibiotics-10-01310-f006:**
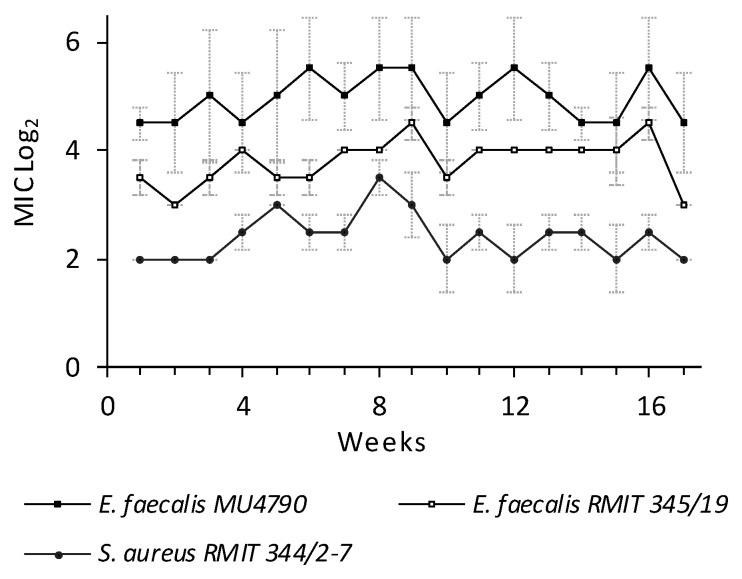
Weekly (W_1_–W_16_) geomean MIC of VRE and MRSA. Strains, serially cultured for 16 weeks in CAMHB with NPBD, subculture to fresh dilution sets, W_1_ to W_15_, and serial passage in CAMHB to W_16–17_. MIC/MBC W_0_ and W_17_ were determined by broth microdilution ([28,29], Appendix A). Inoculum density W_1_ to W_15_ varied <1 log_10._ Assays were repeated twice. MBC ≤ 2× MIC. 2.5% DMSO. Weekly inocula of 1–5 × 10^5^ cfu/mL standardized and verified by viable count.

**Table 1 antibiotics-10-01310-t001:** Antibacterial activity of NPBD against clinically significant species.

Phylum ^a^					
Order (Family)					
	Species	MIC_100_ ^b^	*±*SD	MBC_99.9_ ^b^	*±*SD
Firmicutes/Actinobacteria (Gram-positive)					
Bacillales (Bacillaceae)					
	*Bacillus subtilis* ATCC6633	8	0	16	0
	*B. cereus* RMIT 30/7	9.2	6.6	12.1	4.4
	*Enterococcus faecalis* ATCC 29212 *	16	0	>512	
	*E. faecalis antibiotic resistant* clinical strains (7)	16	0	>512	
	*E. faecium* VRE 345/19-1 & VRE/19	25.4	9.2	>512	
Bacillales (Staphylococcaeae)					
	*Staphylococcus aureus* ATCC 29213 *	4.8	1.9	≥512	
	*S. aureus antibiotic resistant* clinical strains (12)	5.0	1.9	≥512	
	*S. epidermidis* ATCC 35984	5.0	2	≥512	
Lactobacillalles (Streptococcceae)					
	*Streptococcus pneumoniae* ATCC 49619 *	16.0	0	32	0
	*S. pyogenes* ATCC 19615	2.8	1.2	5.7	2.3
	*S. pyogenes* clinical strains (10)	2.7	1	3.4	0.85
Lactobacillalles (Lactobacillaceae)					
	*Lactobacillus casei* RMIT 190/3	49	21.5	338	200
Clostridiales (Peptostreptococcaceae)					
	*Clostridium difficile* ATCC 9689 (*Clostridiodes difficile, Peptoclostridium difficile*)	8	0	16	0
	*C. difficile* (Clinical isolate)	6	2	10	5
Clostridiales (Clostridiaceae)					
	*C. perfringens* NCTC 8237	5	2	10	5
	*C. sporogenes* RMIT 52/4	1	0	1	0
	*C. tetani* RMIT 52/5	1	0	1	0
Corynebacteriales (Corynebacteriaceae)					
	*Corynebacterium xerosis* RMIT53/5	8	0	16	0
	Average	10.5	4.7	281.6	40.9
Proteobacteria (α,β,γ) (Gram-negative, non-enteric, lipo-oligosaccharide)					
Pasteurellales (Pasteurellaceae)					
	*Haemophilus influenzae* (γ) ATCC 49247	0.125	0	0.125	0
	*Pasteurella multocida* (γ) RMIT 284/1-2	2.2	0.8	3.2	1.0
Neisseriales (Neisseriaceae)					
	*Neisseria gonorrhoeae* (β) RMIT 240/2	5	2.3	5	2.3
	*N. gonorrhoeae* WHO strain VI	2	0	2	0
	*N. meningitidis* ATCC 13090	0.5	0	0.5	0
Rhizobiales (Brucellaceae)					
	*Brucella abortus* (α) RMIT 33/1 48 h	42	17	76	32
Pseudomonadales (Moraxellaceae)					
	*Acinetobacter calcoaceticus var. anitratus (*γ) RMIT3131	128	0	256	0
	*Moraxella catarrhalis* (γ) RMIT 211/2	16	0	32	0
	Average	24.5	4.9	46.8	8.9
Proteobacteria (γ,ε) (Gram-negative, enteric, lipo-oligosaccharide)					
Campylobacteriales (Campylobactereaceae)					
	*Campylobacter jejuni* (δ/ε) ATCC 43446 (0:19)	202	74	323	148
	*C. jejuni* NCTC11168	203	66.1	431	128
	*C. jejuni* 54/1-2	203	66	431	128
	*C. jejuni* 331	161	73.9	362	181
	*C. jejuni* antibiotic resistant strain (6)	276	82.79	424	171
	*C. coli*	202	74	406	148
	*C. laridis*	128	0	256	0
	*C. sputorum*	128	0	256	0
	*C. foetus*	203	74	512	0
	*C. hyointestinalis*	128	0	256	0
	Average	183.5	33.73	365.6	75.4
Bacteroidales (Bacteriodaceae)					
	*Bacteroides fragilis* NCTC9343	23	14	39	16
Enterobacterales (Yersiniaceae)					
	*Yersinia enterocolitica (*γ) ATCC 23715	16	0	32	
	*Y. enterocolitica* ATCC 70020	16	0	32	
	Average	18.3	5.90	34.3	14.3
Proteobacteria (γ) (Gram-negative, enteric, lipopolysaccharide)					
Enterobacterales (Morganellaceae)					
	*Proteus mirabilis RMIT 281/1*	180	74	>512 ^c^	
	*P. mirabilis clinical strains (6)*	64	0	>512	
	*P. vulgaris RMIT 281/3*	53	16	181	73
	*P. vulgaris ATCC13315*	8	0	25	9
	Average	76.3	30.44	103.0	36.8
Enterobacterales (Enterobacteriaceae)					
	*Enterobacter aerogenes*	512		≥512 ^c^	
	*Escherichia coli* ATCC 27853	323	132.2	≥512 ^c^	
	*E. coli* ATCC 25922 *	512		>512 ^c^	
	*E. coli* RMIT 1110/1-5 (5)	>512			
	*Klebsiella aerogenes (Areobacter)* ATCC 13048	>512		>512 ^c^	
	*K. oxytoca* RMIT 180/4	≥512		>512 ^c^	
	*K. pneumoniae* ATCC13833	≥512			
	*K. pneumoniae* RMIT 180/2-6	≥512			
	*Salmonella* Typhimurium ATCC 700720	512	0	>512 ^c^	
	*S. enterica* Typhimurium ATCC14028	512		>512 ^c^	
	*Serratia marcescens* RMIT	>512		>512 ^c^	
Pseudomonadales (Pseudomonadaceae)					
	*Pseudomonas aeruginosa* ATCC 27853	>512		>512 ^c^	

^a^ Phylogenetic taxonomy and nomenclature aligns with the National Center for Biotechnology Information (NCBI) for organisms in the public sequence databases. ^b^ MIC_100_ and MBC_99.9_ by CLSI microdilution method (M11-A7; M07-A8) as appropriate for species from a minimum of three independent assays. Ciprofloxacin positive control MIC and MBC were within accepted ranges for QC control strains (*) (Appendix A). Test systems contained 1% *v*/*v* DMSO. ^c^ MBC ≤ 2048 mg/L in presence of ≤5% DMSO. 24 h MIC/MBC except for *Campylobacter* spp. and *Bacteroides* sp. which were 48 h. Geomean data for each species reported with average titres calculated for multiple strains of a species.

**Table 2 antibiotics-10-01310-t002:** Antibacterial activity ^a^ (mg/L) of NPBD in the presence of 1 and 10 molar excess of cysteine and dithiothreitol.

	CAMHB Control	Cysteine (mM)		Dithiothreitol (DTT) (mM)	
		1	10	1	10
*S. aureus* ATCC 29213	4	4	6	11	304
*E. faecalis* ATCC 29212	10	10	23	32	362
*B. subtilis* ATCC 6633	5	7	10	21	362
*P. vulgaris* ATCC 13315	6	8	11	8	215
Mean ± SEM	5.9 ± 2.6	6.8 ± 2.5	11.1 ± 7.3	15.6 ± 11	304 ± 69

^a^ MIC determined by broth microdilution (CLSI, M07-A8) with and without addition of excess cysteine or DTT. No effect on MIC for 0.1 mM. The MIC of ciprofloxacin for *S. aureus* was within QC range (0.25 mg/L).

**Table 3 antibiotics-10-01310-t003:** Antibacterial efficacy of NPBD/antibiotic combinations for *S. aureus* and *E. faecalis*.

		FICI ^a^ and Median ΣFIC ^b^ (Range) for NPBD in Combination with Antibiotics:			
		Erythromycin	Ciprofloxacin	Vancomycin	Tetracycline
*S. aureus*	FICI	1.4 ± 0.4	0.99 ± 0.25	1.15 ± 0.3	0.92 ± 0.25
	ΣFIC_min_	0.88 (0.75–1.02) ^c^	1 (0.27–1.03)	1 (0.56–1.01)	0.76 (0.52–1) ^c^
	ΣFIC_max_	1.38 (1.25–1.5) ^c^	1.25 (1.01–4.5)	1.5 (1.25–4)	1.38 (1.25–1.5) ^c^
*E. faecalis*	FICI	0.75 ± 0.22	1.09 ± 0.16	1.03 ± 0.29	0.51 ± 0.12
	ΣFIC_min_	ND	0.75 (0.56–1)	0.5 (0.38–1.03)	0.19 * (0.19–0.25)
	ΣFIC_max_	ND	1.13 (1.13–2.13)	1.13 * (1.02–1.5)	1.03 * (0.56–1.06)

^a^ Fractional Inhibitory Concentration Index: FICI <0.5 indicates synergy, >0.5 to <4 indicates no interaction, and >4 indicates antagonism; ^b^ ΣFIC_max_ ≥ 2 is proposed as indicative of antagonism, and ΣFICmin <0.75 of synergy; ^c^ N = 2; ***** statistical significance for *E. faecalis* synergism NPBD/Tetracycline 0.00074, and antagonism NPBD/Vancomycin 0.017, NPBD/Tetracycline 0.01 (*p* = 0.05).

## Data Availability

Data is contained within the article or Appendix A.

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
