# Peer review of "Antibacterial Profile of a Microbicidal Agent Targeting Tyrosine Phosphatases and Redox Thiols, Novel Drug Targets"

_antibiotics, 2021, doi:10.3390/antibiotics10111310_

Round 1
Reviewer 1 Report
The authors reported the NPBD as a novel agent targeting a new target. According to their previous work, I speculated that the authors want to expand their findings to support NPBD for further development. The authors have made efforts on chemical or pharmaceutical assays. However, many microbiological tests were not well-designed or well-done according to criteria recommended by CLSI or EUCAST. Please see below for experimental problems I found during my review:
- No need for the average of MIC (Tables 1 & 2).
- The starting inoculum in every time-killing assay seems to be different. Some of them are even up to 107~108 CFU/ml (it should be 106 CFU/ml according to their method section). It’s needed to redo all the experimental to make the data consistent. (Figures 2 & 3)
- The synergistic effect of NPBD and tetracycline is not consistently observed in Table 3 and Figure 5.
- Figure 5. TETRACYCLINE IS NOT A CIDAL AGENT. IT’S STATIC.
- Since the procedure (the inoculum) and statistics were not fit any guideline, it’s no need to show table S7 in this article.
- Table S9. Please make the unit uniform even the values are too high.
- It’s recommended to have an indicated in vivo model if the authors want a conclusion they have in the Abstract.
Also, some format or writing problems as below:
- The introduction should be more concise.
- Supplementary tables should be rechecked for the order.
- Some words are in colors or highlighted. Not sure if any meaning.
- Section 2.1. unit is needed even in brackets.
- It’s acceptable to have results and discussions in the same section, but the writing of this section in this manuscript is so confused and hard to catch the exact data in this study. It’s recommended to rephrase them.
Reviewer 2 Report
The research article entitled "Antibacterial profile of a microbicidal agent targeting tyrosine 2 phosphatases and redox thiols, novel drug targets" by White K et al., evaluated the antibacterial effect of redox thiol oxidation and PTP inhibition and studied their possibilities to use as potential antimicrobials.
Overall, the manuscript is well written, structured and the results are intriguing. As a reviewer considering the general view of readers and peers, I have below questions/comments and also recommended few minor corrections.
Comments:
- The results presented in 2.1 section does not actually reflect the data shown in the Table 1. Especially the MIC titers range in line 120, MIC SD vales in line 122, line 123, line 125, line 127,. Please cross verify the values and provide a detailed information.
- I understand that the authors have tested the antibacterial activity of NPBP against clinically significant species but it would be interesting also to check if these have any effect on mycoplasmas?
- Secondly, is there any negative and positive control for the data presented in the Table .1 ? if not why it is ignored?
- One of the other major concern I have is did the authors determine the statistical significance of these data presented in Figure 2, 3, 4 and 5? If not, please do so and provide the P-values of the data significance in the figures
- Lastly, the authors have stated 4 patents as a part of this study, when looked out for the authors in those patents the first three patents does not have any of the authors involved in this study. Did the authors acquire permission to use these three patents? If so, please mention the same in the acknowledgements section.
- Minor changes: Provide reference citations for the sentences ending in lines 30, 36, 139 and 96
- Minor: Correct the spelling of enzymic to "enzymatic" through out the manuscript for example in lines 77,78, 82 etc
- Line 103: Correct to Protozoan
- Line 204: Correct to Pharmacodynamics
- Line 336: Change to "more significant than the difference"
- Line 395: Correct to "Physicochemical"
Reviewer 3 Report
Article is good corrected. Authors present activity of protein tyrosine phosphatase (PTP) inhibitor and redox thiol oxidant, nitropropenylbenzodioxole (NPBD). Studies were performed against 39 bacterial species, using many microbiological methods. Methodology is properly described. This article is important because presents possibility of antibacterial using of PTP.
Author Response
The author's thank reviewer 3 for their comment. No suggestions for revision were provided.
Reviewer 4 Report
In this manuscript, the authors studied the effect of NPBD, a potential inhibitor of phosphoprotein phosphatase (PTP) and redox thiol oxydant on a wide panel of bacteria. They found bactericid effect of NPBD on diverse bacteria, low cross reactivity with other antibiotics and no induction of resistance for antibiotic-resistant species. Besides, they determine the toxicity and tolerance of NPBD in rat and mice and studied bioavaibility of NPBD in presence of albumine and bv varying redox potential. The experiments are well conducted and the main conclusion is that PTP are attractive targets for oxydant drugs such as NPBD.
My main concern is about the roles of PTP in bacterial physiology, adaptation to stress and virulence. PTP have been shown to be important for virulence and biofilm formation. Though the scope of the authors is to evaluate how targeting PTP may influence bacterial viability, they also make reference to function of PTP in virulence. These two functions should be clearly separated, with bacterial intracellular function necessary for “normal” growth on one site and virulence and adaptation to stress (which is not evaluated her but should be done in a separated study) on the other side. In particular the effect of NPBD on bacteria-host relationship during infection, biofilm formation should be more clearly discussed
Last paragraph of section 2.1 (lines 187-218) should be written in a clearer way, separating inner function of PTP and PTP as virulence factor, either secreted of required for attachment, polysaccharide production, biofilm formation. This paragraph is not well ordered and is as a catalog of all known PTP. Lines 164-167 should be connected here. Last paragraph of table S10 should be inserted here.
Minor points
- Why NPBD kills P. vulgaris in CAMHB but not in water. Does it require cation. This point should be more clearky discussed.
- line 161: Candida albicans is a budding yeast. It appears in a list of bacteria and this may be confusing.
- In supplementary data 4, Results summary Part D, there is a repetition of “at” “Mortality at At 2000mg/kg/day...”
Author Response
Response to Reviewer 4, Round 2
General comments
The functions of PTP in stress responses and virulence activities, and particularly bacteria-host relationships, are not adequately addressed
‘Virulence factors’ are descriptors for very diverse bacterial activities that cause tissue damage and activities that subvert and suppress host-specific stress responses. Signaling molecules involved in infection include tyrosine phosphatases, quorum sensing auto-inducers and the unique bacterial second messenger, cyclic di-guanosine monophosphate, all of which can interact and play context-dependent positive and negative roles in microbial stress survival and disease-causing activities.
The important role PTP play in bacterial infectivity, however, is outside the scope of this paper. We address the in-vitro effects of NPBD on bacterial virulence activities in a companion paper in preparation which will address some of the concerns raised here. Investigating bacterial activities which directly affect host stress responses is complicated. Many of the required cell and animal assays are undertaken in the pre-clinical evaluation of a developmental agent.
Last paragraph of section 2.1 (lines 187-218) should be written in a clearer way.
We have added to (lines 190-195; lines 225-230) and reorganised parts of the section reporting characterised PTP functions and hope that it is now clearer. It is difficult to avoid a litany of results with no overall connectivity because the functions of PTPs are as yet in early stage investigation and data has evolved piecemeal in the literature. As more PTPs are characterised, a more coordinated understanding of these complex interactions will emerge.
Specific comments
Last paragraph of Table S10 should be inserted [in article text].
This is reported data and as we have no additional data we hesitate to make additional comment in the text, leaving our suppositions as a comment on the data presented in Table S10.
Why NPBD kills P. vulgaris in CMHB but not in water? Does it require cation?
Nitroalkene benzenes are strongly electrophilic and react strongly with anions, e.g. thiolates. NPBD is non-ionic and permeates cells readily. De-ionised water was used but any cations present would not interfere with NPBD chemistry or ability to enter cells.
- albicans is a budding yeast. It appears in a list of bacteria and this may be confusing.
The inclusion of significant urogenital pathogens illustrates the broad antimicrobial activity of NPBD and supports the proposal that it would be a suitable agent for treatment of certain types of mucosal infections.
In supplementary data 4, results summary Part D, there is a repetition of “at”: “Mortality at AT 2000mg/kg/day”.
The typo has been corrected.
General response
We hope the changes we have made deal adequately with the concerns raised by the Reviewer while remaining within the rather narrow scope of this article, and meeting journal length requirements for publication. Several of the issues raised will be dealt with in additional publications currently in preparation on bacterial virulence factors and antifungal activity.
Round 2
Reviewer 1 Report
Authors obviously don't want to revise their article.
Just a simple example: "the criteria of CLSI"
- no average MIC was determined (or needed) in CLSI guidelines.
- according to CLSI guidelines, the starting inoculum is 5*10^5 CFU. it's not acceptable to have only 10^4 CFU/ml (Table S8). It's good to know the start inoculum which is also a QC for the BMD test.
Also in my previous general comment: "According to their previous work, I speculated that the authors want to expend their finding to support NPBD for further development." This article is a me-too study (no novelty). Thus, the article is not qualified to publish in this journal.
